# Intracellular Inclusions Induced by Patient-Derived and Amplified α-Synuclein Aggregates Are Morphologically Indistinguishable

**DOI:** 10.3390/cells14100684

**Published:** 2025-05-09

**Authors:** Rabab Al-Lahham, Mark E. Corkins, Mohd Ishtikhar, Prakruti Rabadia, Santiago Ramirez, Victor Banerjee, Mohammad Shahnawaz

**Affiliations:** 1Mitchell Center for Alzheimer’s Disease and Related Brain Disorders, Department of Neurology, McGovern Medical School, The University of Texas Health Science Center at Houston, Houston, TX 77030, USA; mohd.ishtikhar@uth.tmc.edu (M.I.); prakrutirabadia@gmail.com (P.R.); santiago.d.ramirez@uth.tmc.edu (S.R.);; 2Department of Pediatrics, McGovern Medical School, The University of Texas Health Science Center at Houston, Houston, TX 77030, USA; mec477@pitt.edu

**Keywords:** Parkinson’s disease, multiple system atrophy, α-synuclein, aggregation, SAA, biosensor cells

## Abstract

Lewy Body Disease (LBD) and Multiple System Atrophy (MSA) are synucleinopathies with distinct prognoses and neuropathologies, however, with overlapping clinical symptoms. Different disease characteristics are proposed to be determined by distinct conformations of alpha-synuclein (α-Syn) aggregates, which can self-propagate and spread between cells via a prion-like mechanism. The goal of this study is to investigate whether α-syn aggregates amplified from brain and CSF samples of LBD and MSA patients using the Seed Amplification Assay (SAA) maintain α-Syn seeding properties similar to those of α-syn aggregates derived from patients’ brains. To address this, SAA-amplified and un-amplified α-Syn aggregates from LBD and MSA patients’ brains, as well as SAA-amplified α-Syn aggregates from LBD and MSA patients’ CSF samples, were used to treat synuclein biosensor cells, and induced intracellular α-Syn inclusions were analyzed by confocal microscopy. Our data indicate that induced α-Syn aggregates from LBD and MSA patients’ brains have similar seeding properties and morphological characteristics in the α-Syn biosensor cells as those amplified from LBD and MSA patients’ brains, as well as those amplified from LBD and MSA patients’ CSF samples. In this study, we demonstrated that, regardless of the source of aggregates, the seeds from LBD and MSA produce cellular accumulation of α-Syn with distinct morphologies, confirming the presence of different conformational strains of α-Syn in LBD and MSA and allowing us to differentiate synucleinopathies based on the morphology of aggregates and seeding properties.

## 1. Introduction

Synucleinopathies are neurodegenerative diseases identified by the deposition of α-Synuclein (α-Syn) aggregates as their histopathological hallmark [1,2,3,4,5]. They include Lewy body diseases (LBDs), such as Parkinson’s disease (PD), PD with Dementia (PDD), and dementia with Lewy bodies (DLB), and multiple system atrophy (MSA) [1,6,7]. Although they share overlapping clinical symptoms, their prognoses and neuropathological features differ significantly, which makes early diagnosis particularly challenging, especially in the initial stages of the disease. α-Syn misfolds and aggregates, producing Lewy Bodies (LBs) as intra-neuronal inclusions in PD, PDD, and DLB, while they deposit as Glial Cytoplasmic Inclusions (GCI) in glial cells in MSA. Researchers have proposed that the individual disease criteria are based on the distinct conformations of alpha-synuclein aggregates, which can self-propagate between cells and spread via a prion-like mechanism, thus advancing the neuropathological progression of the disease [8,9,10,11,12,13,14,15,16,17,18,19,20,21,22,23,24]. Cell-to-cell propagation of α-Syn aggregates has been demonstrated in vitro in multiple cell lines as well as in neurons grafted into PD patients’ brains [8,9,11,12,13,14,23,25]. Similarly, intra-cerebral or peripheral injection of α-Syn inclusions into mice induced propagation of α-Syn aggregates [17,19,20,23,26,27,28,29,30,31,32,33,34,35].

The Seed Amplification Assay (SAA), previously known as Protein Misfolding Cyclic Amplification (PMCA), is an established method to detect prions. We adapted this assay to detect α-Syn aggregates in the CSF of PD patients in a reliable and reproducible fashion with high sensitivity and specificity [36,37]. Recently, we reported that the SAA could differentiate CSF-derived α-Syn aggregates in PD from those in MSA based on the conformation of the aggregates. This suggests that α-Syn aggregates in PD and MSA have different conformational strains [38]. Later, these data were confirmed for oligomeric α-Syn at early stages of the disease [39]. Recent studies have shown that SAA-amplified α-Syn aggregates have different structures to those in the brain, suggesting the presence of other factors that play a role in the propagation of the seeds [40]. However, whether SAA amplification affects the seeding properties of α-Syn aggregates in cells is yet to be determined. Recent studies have demonstrated that α-Syn spreading in a prion-like manner precedes neuropathology, which is the gold standard for disease diagnosis [41]. Therefore, it is crucial to investigate and test the seeding of α-Syn in an in-vitro system that will assist in early diagnosis, as well as to evaluate therapeutic interventions.

In the present study, we investigated whether SAA-amplified α-Syn aggregates from LBD and MSA patients’ brains maintain their seeding properties using synuclein biosensor cells, and whether α-Syn aggregates present in patients’ CSF have similar seeding properties and morphological characteristics to those in the brain. We report an important seeding characteristic of α-Syn aggregates and show how, regardless of the source of aggregates, LBD and MSA samples produce cellular accumulations of α-Syn with different morphologies, confirming the presence of different conformational strains of α-Syn in LBD and MSA. Our findings also demonstrate that the seeding properties of SAA-amplified aggregates are maintained after amplification and that CSF-induced α-Syn inclusions are similar to those induced with brain samples.

## 2. Materials and Methods

### 2.1. Patient Samples

Brain tissue samples from three LBD and three MSA patients were supplied by Banner Sun Health Research Institute, Sun City, AZ, USA and Mayo Clinic, Rochester, MN, USA. Brain tissue samples from three healthy control (HC) subjects were obtained from NDRI (National Human Tissue Resource Center). Brain tissue frozen samples were processed as described previously [38]. Human samples were handled according to the universal precautions for working with human specimens, as regulated by the Institutional Review Board of The University of Texas Health Science Center at Houston (HSC-MS-14-0608).

Cerebrospinal fluid (CSF) samples were derived from three patients clinically diagnosed with LBD, three with MSA, and three controls collected from individuals affected with neurological diseases other than LBD or MSA. Table 1 shows an overview of the patients’ demographic characteristics. Samples were supplied by the Mayo Clinic. LBD and MSA clinical diagnoses were made in accordance with internationally standardized criteria, involving the UK Brain Bank guidelines. CSF samples were processed as published previously [38]. CSF collection methods were accepted by the institutional review boards at the supplier study centers, and all participants issued written informed consent.

### 2.2. Preparation of Postmortem Brain Tissue Insoluble Fractions

To prepare insoluble fractions, brain homogenates were centrifuged at 100,000× *g* for 30 min at 4 °C. Pellets were resuspended in an equal volume of PBS with added protease inhibitors and were used for treatments as the insoluble fraction, using 80 µg total protein per well of the 8-well chambered cover glass.

### 2.3. Preparation of α-Syn Monomer

Purification and identification of α-Syn monomer was implemented as previously described [37,38]. Total protein concentration was measured using BCA protein assay kit with bovine serum albumin as a protein standard (Pierce # 23225, ThermoFisher Scientific, Waltham, MA, USA). Purity of the α-Syn protein was examined using silver staining.

### 2.4. α-Syn Seed Amplification Assay

αSyn-SAA was performed as previously described [37,38]. Briefly, seed-free α-Syn samples (1 mg/mL), in 100 mM PIPES, pH 6.5, 500 mM NaCl, were added with 5 μM ThT to 96-well opaque plates (Costar, Washington, DC, USA, # 3916) up to 200 μL. A 40 μL sample of CSF or brain tissue homogenate (final concentration of 0.001%) from patients as well as controls were added per test. Positive controls included healthy subjects’ CSF samples that were spiked with preformed α-Syn oligomeric seeds. Samples were exposed to cyclic agitation (1 min at 500 rpm with shaking and then 29 min with no shaking) at 37 °C. ThT fluorescence was measured periodically (excitation/emission at 435/485 nm) with a microplate spectrofluorometer (Gemini-EM, Molecular Devices, Sunnyvale, CA, USA).

The second rounds of the SAA were conducted in triplicates by diluting 100 times the amplified α-Syn from a single well of the first round using fresh α-Syn monomer. The assay was repeated two successive times to acquire materials equivalent to the second rounds of amplification. The SAA conducted with the patients’ samples (CSF or brain homogenate) was considered the first round of amplification. Samples from the second round of amplification were used for proteinase K digestion and cell treatment assays.

### 2.5. Proteinase K Digestion of α-Syn Aggregates Amplified by SAA

Brain homogenate and CSF samples containing α-syn aggregates amplified by SAA were treated with 1 mg/mL concentrations of proteinase K at 37 °C for 1 h. The reaction was stopped by heating the sample in NuPAGE LDS buffer at 95 °C for 10 min. The digested products were resolved by 12% Bis-Tris gels (Invitrogen, Waltham, MA, USA). Proteins were electrophoretically transferred to nitrocellulose membranes (Amersham Biosciences, Piscataway, NJ, USA). Membranes were blocked with 5% *w*/*v* non-fat dry milk in PBS–Tween 20 (PBS (Hyclone SH.30258.02, pH 7.2, 0.1% (*v*/*v*) Tween 20) at room temperature for 1 h. After blocking, the antibodies against α-syn, i.e., anti-α-syn clone 42 (BD Biosciences, Franklin Lakes, NJ, USA), was used which is raised against the middle region (residues 15–123) of the α-syn. The blots were developed using ECL prime detection Western blotting reagents (Amersham Biosciences).

### 2.6. Negative Stain Electron Microscopy Imaging

The samples were prepared following the method previously described [42]. In brief, 5 µL of each sample was applied to glow-discharged carbon-coated copper grids and allowed to adsorb for 30 s at room temperature. Excess liquid was carefully removed using filter paper. The grids were then washed by gently touching them to a drop of MilliQ water, followed by blotting; this washing step was repeated twice. Subsequently, 5 µL of 2% (*w*/*v*) uranyl acetate was applied for 22 s for negative staining. After staining, excess stain was blotted off, and this process was repeated three times. Finally, the grids were allowed to air-dry thoroughly at room temperature. Imaging was performed using a JEOL JEM-1400 transmission electron microscope (JEOL USA, Inc., Peabody, MA, USA) operated at an accelerating voltage of 120 kV.

### 2.7. Synuclein Biosensor Cells

HEK293T cell line stably expressing α-Syn (A53T)-CFP/YFP fusion proteins was generated by Marc Diamond group as described previously [43]. At baseline, the α-Syn proteins exist in a stable, soluble form throughout the cell, while exposure to exogenous α-Syn seeds from LBD and MSA patients’ samples leads to α-Syn aggregation, thereby bringing CFP and YFP in close proximity and thus generating a FRET signal that can be imaged by confocal microscopy.

### 2.8. FRET Seeding Assay

Biosensor cells were plated in an 8-well chambered cover glass (Nunc Lab-Tek Chambered Coverglass, Thermo Scientific, Waltham, MA, USA) at 3000 cells/well and incubated overnight. The next day, insoluble fractions from brain tissue homogenates were sonicated for 90 s at 30 A to break any big fibrils into smaller shorter fibrils and oligomers. The SAA-amplified brain fractions were sonicated for 60 s at 30 A, while SAA-amplified CSF samples were sonicated for 30 s at 30 A. The SAA-amplified samples were used at 400 nM concentration in 300 µL DMEM high glucose media (Sigma-Aldrich, St. Louis, MO, USA) supplemented with 10% fetal bovine serum (FBS) (Gibco, Grand Island, NY, USA), GlutaMax (Gibco, Grand Island, NY, USA), penicillin (100 U/mL), and streptomycin (100 µg/mL) (Gibco, Grand Island, NY, USA). Un-amplified brain homogenates insoluble fractions from LBD and MSA patients were used at 80 µg total protein per well. Cells were imaged using confocal microscopy at 3-, 10- and 13-days post-treatment.

### 2.9. Confocal Microscopy Imaging

Zeiss LSM800 (Oberkochen, Germany) equipped with an Airyscan detector was used to image cells. Channels were set up as follows: CFP donor (Ex 405/Em 400–500 nm), YFP acceptor (Ex 488/Em 530–600 nm), and FRET signal (Ex 405/Em 530–600 nm). Images were imported to ImageJ 1.54 s software (NIH) for analysis. The FRET signal area was used to determine the aggregate load, expressed as burden percentage of total cell area.

### 2.10. Statistical Analysis

Quantification of aggregate load was carried out using 3 images per patient. Data are presented as mean ± SEM for 3 patients. All data were analyzed using GraphPad prism 9.0 software (GraphPad Software Inc., La Jolla, CA, USA). Statistical analysis was carried out as indicated in the figure legend. Results were considered significant if the *p* value was <0.05.

## 3. Results

### 3.1. Amplification of α-Syn Aggregates from Patients’ Brains and CSF Samples and Their Characterization

We amplified α-syn aggregates from brain and CSF samples of LBD and MSA patients using α-syn-SAA, formerly known as α-syn-PMCA. In parallel, amplified brain and CSF samples from healthy individuals were used as controls. Consistent with our previous results, the maximum fluorescence and the kinetics of aggregation, as monitored by ThT fluorescence, of LBD CSF samples were distinct from CSF samples of MSA patients [38]. The Maximum ThT fluorescence of MSA-amplified α-syn aggregates was always below 1000. However, the LBD-amplified α-syn aggregates gave higher ThT fluorescence signals >2000. (Figure 1A right panel). The results were similar with LBD and MSA patients’ brains (Figure 1A left panel). None of the brain and CSF samples from healthy individuals gave higher ThT fluorescence above background levels (Figure 1A left and right panels).

To confirm whether α-syn aggregates amplified from patients’ brains and CSF samples contained strain characteristics, we performed limited proteolytic digestion with proteinase K. For that purpose, we used aggregates amplified from the second round of SAA. Similar to what we reported before, the amplified aggregates from brain as well as CSF samples of LBD and MSA patients retained strain characteristics [38]. The profiles of protease-resistant fragments of aggregates amplified from LBD patients differed from those amplified from MSA patients regardless of the origin of samples (Figure 1B). The aggregates from LBD patients’ samples gave four protease-resistant fragments ranging from 4 to 10 kDa, whereas aggregates amplified from MSA patients’ samples resulted into 1–2 bands (4 to 6 kDa) (Figure 1B).

To confirm the presence of filaments, both SAA-amplified samples and un-amplified healthy control samples were imaged using a JEOL JEM-1400 transmission electron microscope operated at 120 kV, at a nominal magnification of 120k (Figure 1C). Both LBD and MSA amplified filaments exhibited an average width of approximately 10 nm consistent with our previous report [38] (Figure 1C). Due to the resolution limitations of negative stain electron microscopy, structural distinctions between LBD and MSA filaments could not be resolved. No filamentous structures were detected in the control samples, confirming the specificity of filament formation in the amplified reactions (Figure 1C).

### 3.2. Brain Homogenates of LBD and MSA Patients Induce Intracellular α-Syn Inclusions with Distinct Morphologies

We treated α-Syn biosensor cells with brain homogenates’ insoluble fractions from LBD and MSA patients and studied the formation of inclusions via confocal microscopy. As shown in Figure 2, exposure to LBD and MSA brain homogenate insoluble fractions produced inclusions that show YFP and CFP, as well as a FRET signal, indicating the aggregation of α-Syn reporter proteins, therefore, transferring energy and generating a FRET signal that can be imaged. We could also show that MSA α-Syn seeds induce formation of inclusions with unique morphologies compared to LBD α-Syn seeds. MSA-induced inclusions looked filamentous and threadlike (wispy as previously referred to by Marc Diamond’s group) all through the cytoplasm of the cells, while LBD-induced inclusions looked punctate and dotted (as described by Marc Diamond’s group) [25]. These results support the presence of different structures of α-Syn fibrils in these synucleinopathies. In contrast, treatment of cells with brain homogenate insoluble fractions from control subjects did not produce any inclusions.

### 3.3. Morphologies of α-Syn Inclusions Induced with Un-Amplified and SAA-Amplified LBD and MSA Brain Homogenates Are Similar

Exposure of α-Syn biosensor cells to the insoluble fractions of LBD and MSA patients’ brain homogenates induced the formation of distinct morphological inclusions. To investigate whether amplified α-Syn seeds maintain these characteristics, we analyzed the morphologies of inclusions formed after exposure of the cells to SAA-amplified α-Syn seeds from LBD and MSA brain samples, and we found similar morphological characteristics with amplified α-Syn seeds as those with un-amplified samples (Figure 3). Furthermore, MSA-seeded cells had large filamentous inclusions, while LBD-seeded cells had smaller and more punctate inclusions. In accordance with a previous study [25], our data support that LBD and MSA have distinct α-Syn conformational strains, whether amplified or un-amplified.

### 3.4. SAA-Amplified LBD and MSA CSF Samples Induce the Formation of α-Syn Aggregates with Similar Morphologies to SAA-Amplified and Un-Amplified Brain Samples

To investigate if CSF-derived α-Syn seeds are representative of brain-derived seeds, we treated α-Syn biosensor cells with SAA-amplified α-Syn seeds from LBD and MSA patients’ CSF samples from the second cycle of amplification. The use of second-cycle amplified samples results in more aggregates and less interference from other factors in the CSF. We observed the formation of inclusions with morphologies similar to those formed after exposure of the cells to SAA-amplified and un-amplified α-Syn seeds from LBD and MSA patient brain samples (Figure 4), whereas treatment of cells with amplified α-Syn seeds from control subject CSF samples did not produce any inclusions.

### 3.5. MSA Samples Are More Seeding-Competent than LBD Samples

Further, we investigated whether the seeding properties of α-Syn aggregates from LBD patients are similar to α-Syn aggregates from MSA patients by following the same cells for 2 weeks. Treatment of α-Syn biosensor cells with MSA samples produced bigger and a significantly higher number of α-Syn inclusions as compared to LBD samples, whether they were amplified or un-amplified, at both 10 and13 days post-treatment (Figure 5A,B). We found that α-Syn aggregates from MSA samples are more seeding-competent than α-Syn seeds from LBD samples, where there were many α-Syn inclusions seen in cells exposed to MSA-derived aggregates as compared to LBD-derived aggregates. More cells had MSA-induced inclusions, and the inclusions took over more of the cytoplasmic space like a cotton gauze net. Quantification of aggregate burden using the FRET signal relative to area covered by cells confirmed this observation (Figure 5C).

## 4. Discussion

Although recent studies have demonstrated the presence of different strains of α-Syn characterizing different synucleinopathies, like PD and MSA [8,23,25,38,44,45,46], the propagation of these strains in cellular or animal models, as well as strain-specific diagnosis and therapy, is still understudied. Studies have shown that the structures of brain-derived α-Syn filaments differ from those formed using recombinant α-Syn, suggesting that using recombinant proteins to study protein aggregation and propagation might not represent what happens in the brain [44,47,48]. Furthermore, there is debate among researchers if the SAA replicates the structure of α-Syn filaments during amplification [40]. In this study, we used FRET biosensor cells that stably express α-Syn CFP/YFP carrying the SNCA A53T mutation to investigate the seeding activity of amplified vs un-amplified insoluble fractions from LBD and MSA patients’ brains. We provide here the first direct demonstration, to our knowledge, that SAA-amplified α-Syn aggregates from LBD and MSA patients’ brains have similar seeding properties and morphological characteristics of inclusions to those of un-amplified aggregates as well as SAA-amplified aggregates from CSF samples. These results suggest that our SAA method does not alter the seeding activity (or conformation) of α-Syn aggregates in-vitro. Furthermore, these data further support that the CSF-derived α-Syn aggregates reflect similar characteristics to those derived from the brain and, therefore, provide an aid for clinical diagnosis. We found that SAA-amplified α-Syn aggregates from MSA patients’ CSF and brain samples have different morphological features than LBD samples, suggesting again that they are different conformational strains [38]. These observations are in accordance with a previous study by Marc Diamond’s group that showed distinct α-Syn seed characteristics in PD and MSA [25], suggesting distinct α-Syn strains. It is worth noting that previous studies used lipofectamine to increase the uptake of α-Syn into the cells [25]; however, to avoid any unwanted effects of lipofectamine on the health and viability of cells, we did not use lipofectamine in our seeding assays, which would be more representative of α-Syn uptake in-vivo. Another important aspect of our results is that we used all channels of confocal microscopy to ensure that the aggregates we saw were indeed real aggregates, and not due to auto fluorescence of cells, by looking at the FRET signal which can be generated only after CFP and YFP come in close proximity due to α-Syn monomer aggregation. Of note is that here we sonicated the fibrils, whether the BH-amplified, CSF-amplified or the un-amplified ones. Sonication has been shown to produce shorter filaments with more potent seeding activities [49]. It has also been shown that short filaments exhibit the most seed-potent properties [16,50].

By studying the time course of aggregation caused by α-Syn aggregates from brains or CSF samples, we show that MSA α-Syn aggregates are more seeding-competent than LBD aggregates, which is consistent with our previous finding that MSA patients’ CSF samples aggregate faster in the SAA than PD samples [38]. This higher seeding capability was seen in both SAA-amplified as well as un-amplified samples, indicating that the SAA does not change the seeding properties of α-Syn aggregates. This is also consistent with previous reports showing lower seeding potency associated with considerably fewer and smaller aggregates induced by PD compared to MSA [8,16]. Even though some of the previous studies showed α-Syn seeding activity of MSA samples in-vitro and in-vivo (but not for PD samples), our data show seeding activity of LBD samples, whether they were from patients’ brains or CSF, which is consistent with other studies [16,25].

Both PD and MSA are very devastating and costly diseases, with the incidence of both projecting to increase as the population is aging, thus making them major health concerns. Our present work demonstrates an important role for the α-Syn seeding assay in discriminating between LBD and MSA, thus supporting the SAA method used for diagnosis. In addition, we revealed that α-Syn seeds present in patients’ CSF samples reflect those present in the brain, therefore confirming that the use of CSF as a diagnostic route would improve early diagnosis and therapy for patients.

In summary, the present study supports the idea that the SAA can be used to amplify misfolded protein aggregates in body fluids and tissues that cannot be, otherwise, detected due to low concentrations. A growing body of evidence supports the use of the SAA as an early diagnostic tool to differentiate between LBD and MSA; therefore, this study provides a seeding assay that confirms the value of the SAA. Furthermore, the present study also provides evidence for a new diagnostic tool that, together with the SAA, will provide a stronger, more affirmatory, diagnosis. Further studies include extending these methods towards the use of un-amplified CSF for seeding.

Finally, the findings of this study are critical as they suggest that seeding activity assays can be used in conjunction with the SAA, or other methods implemented in the field for detection of misfolded protein aggregates, as screening tools for the presence of protein aggregates seeds. These assays, therefore, play a crucial role in understanding the pathogenesis of synucleinopathies, especially at early stages of the disease, and also pave the way for novel therapeutic strategies that will benefit both PD as well as MSA. Marc Diamond and his group have already shown that the seeding activity of proteins might occur much earlier than the formation of detectable aggregates by IHC, the standard gold method for definite diagnosis of postmortem tissues. Availability of seeding assays to test seeding-competent proteins is an important additional diagnostic test that will help with testing different pharmacological agents for the development of therapeutic interventions.

## Figures and Tables

**Figure 1 cells-14-00684-f001:**
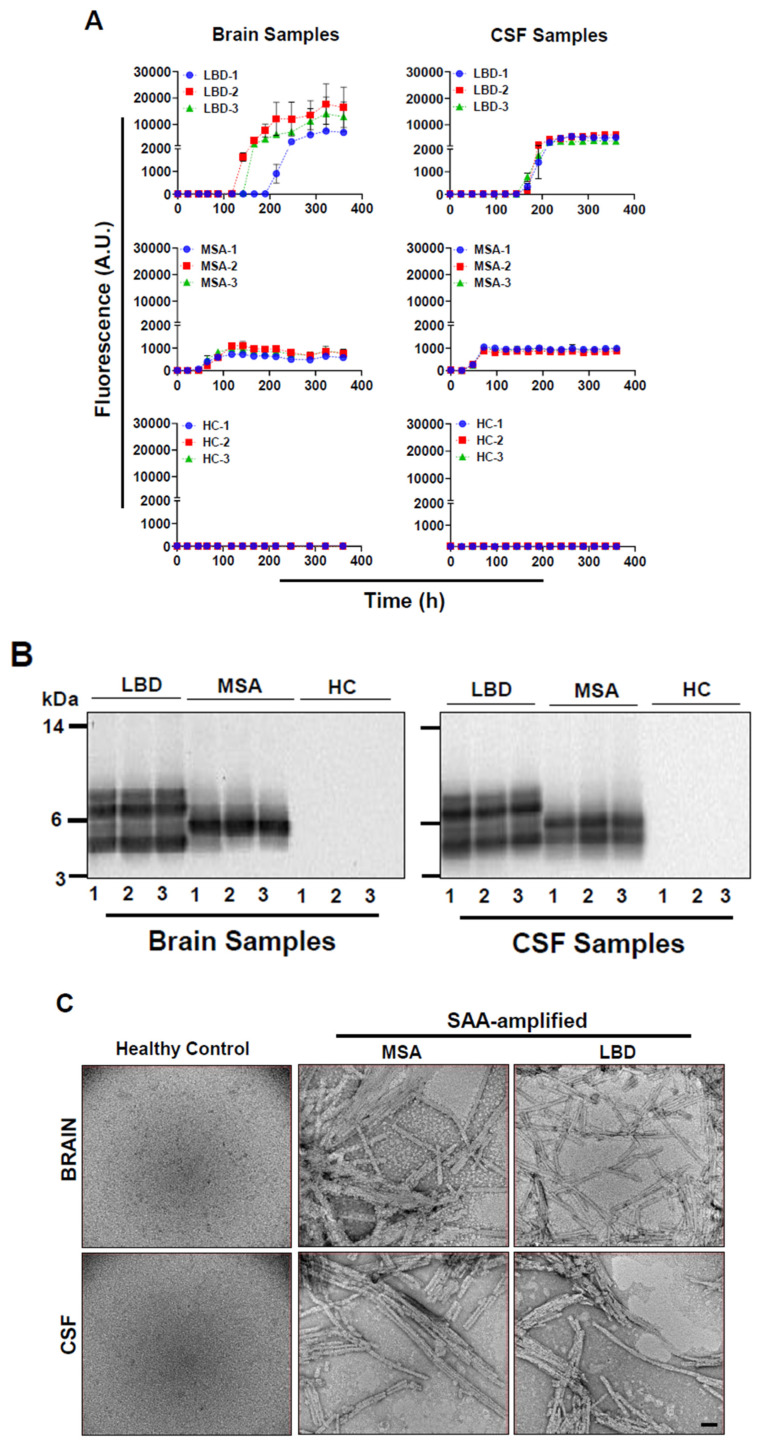
Amplification of α-syn aggregates from patients’ brains and CSF samples and their characterization. Brain homogenate (0.001%) or CSF sample (40 µL) was added to the wells of a 96-well plate, and α-Syn-SAA was started by adding α-Syn monomers and thioflavin-T. The extent of aggregation was monitored by the increase in fluorescence. (**A**): The representative aggregation profiles of brain samples from Lewy Body Disease (LBD; n = 3), Multiple System Atrophy (MSA; n = 3) and Healthy controls (HC; n = 3) (Left Panel); and aggregation profiles of CSF samples from LBD (n = 3), MSA (n = 3), and HC (n = 3) (Right Panel) are shown. Each aggregation curve represents an individual biological sample performed in triplicate. Error bars indicate the standard error of the mean (SEM). (**B**): Amplified α-syn aggregates (0.5 mg/mL) were incubated with Proteinase K (PK; 1 mg/mL) at 37 °C for one h. Proteins were separated on a 12% Bis-Tris gel and immunoblotted with anti-alpha synuclein antibody, BD Biosciences. The profiles of protease-resistant fragments of amplified aggregates from brain samples of LBD (n = 3) and MSA (n = 3) patients, as well as CSF samples of LBD (n = 3) and MSA (n = 3) patients, are illustrated. The complete digestion of SAA end products from brain and CSF samples of HC (n = 3) is shown. (**C**): Representative negative stain electron micrographs of SAA-amplified filaments from MSA and LBD samples, alongside un-amplified healthy control samples, imaged using a JEOL JEM-1400 transmission electron microscope. The scale bar represents 100 nm.

**Figure 2 cells-14-00684-f002:**
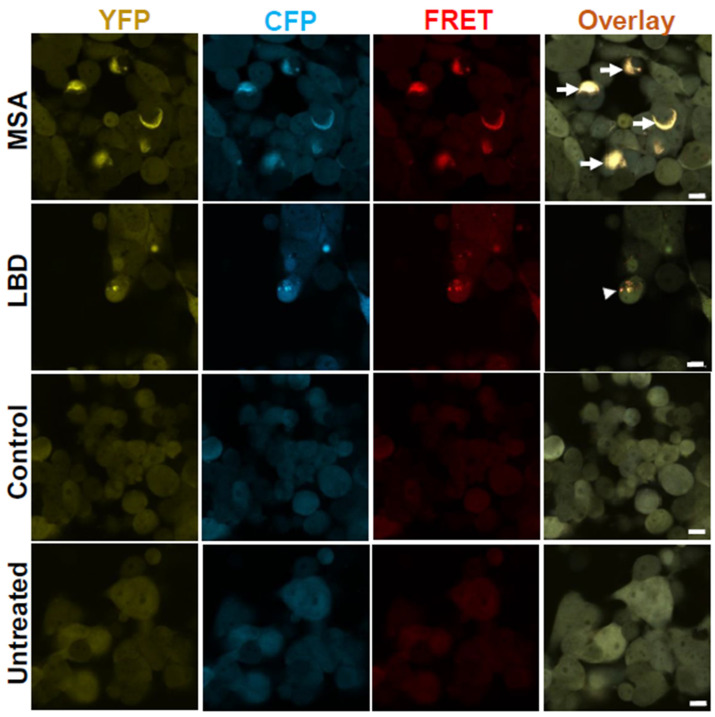
α-Syn aggregates from LBD and MSA patients’ brains seed inclusions with distinct morphologies. Confocal images from α-Syn biosensor cells seeded with insoluble fractions from MSA and LBD patients’ brain homogenates show filamentous threadlike (arrows) and punctate dotted (arrow heads) inclusions, respectively. Cells seeded with insoluble fraction from healthy controls brain homogenates as well as untreated cells do not show any inclusions. Images were taken 10 days post-treatment. Scale bars are 10 µm.

**Figure 3 cells-14-00684-f003:**
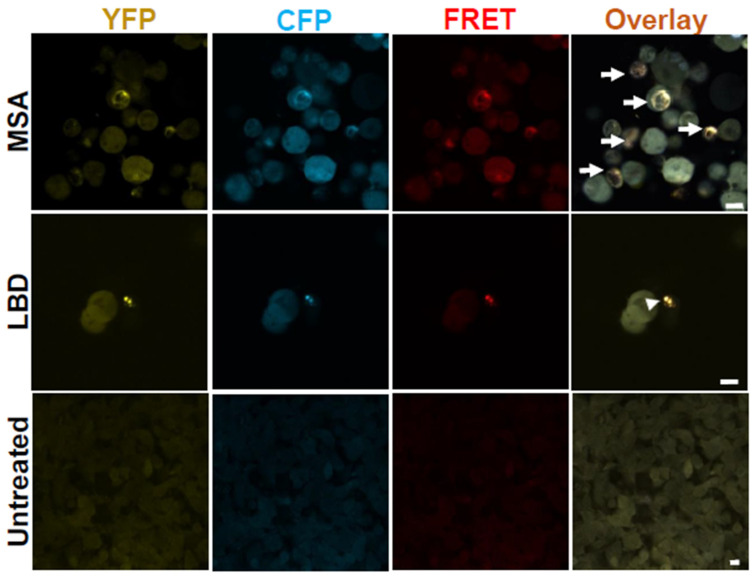
SAA-amplified α-Syn aggregates from LBD and MSA patients’ brains seed inclusions with morphologies similar to those of un-amplified aggregates. Confocal images from α-Syn biosensor cells seeded with SAA-MSA and SAA-LBD patients’ brain homogenates show filamentous threadlike (arrows) and punctate dotted (arrow heads) inclusions, respectively. Untreated cells do not show any inclusions. Images were taken 72 h post-treatment. Scale bars are 10 µm.

**Figure 4 cells-14-00684-f004:**
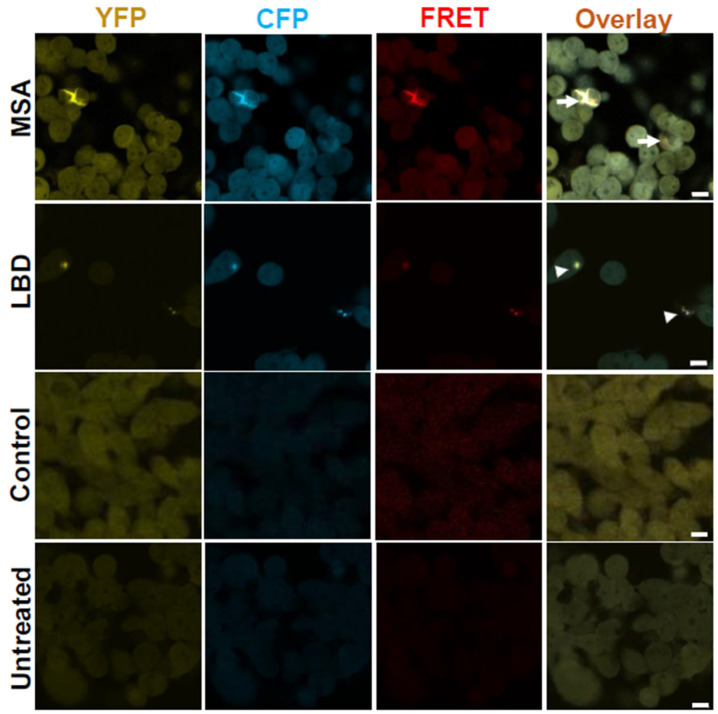
α-Syn aggregates from LBD and MSA patients’ CSF seed inclusions with distinct morphologies similar to SAA-amplified and un-amplified samples from LBD and MSA patients’ brains. Confocal images from α-Syn biosensor cells seeded with SAA-MSA and SAA-LBD patients’ CSF show filamentous threadlike (arrows) and punctate dotted (arrow heads) inclusions, respectively. Cells seeded with SAA-HC CSF as well as untreated cells do not show any inclusions. Images were taken 72 h post-treatment. Scale bars are 10 µm.

**Figure 5 cells-14-00684-f005:**
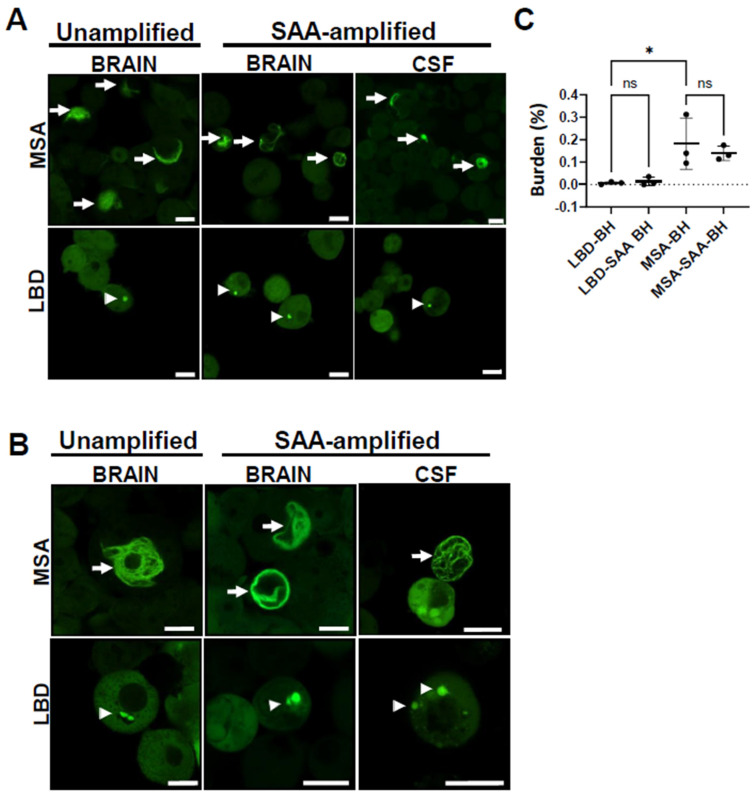
MSA-derived seeds are more seeding-competent than LBD-derived ones. (**A**) Un-amplified and SAA-amplified samples from MSA and LBD patients’ brains show aggregate morphologies similar to those of SAA-MSA and SAA-LBD from patients’ CSF samples. Images were taken 10 days post-treatment for un-amplified brain homogenates samples, and 72 h for amplified samples from brain homogenates and CSF. (**B**) Un-amplified and SAA-amplified samples from MSA and LBD patients’ brains as well as SAA-MSA and SAA-LBD from patients’ CSF samples, show consistent morphologies after 13 days of treatment. Arrows show filamentous MSA-seeded inclusions, while arrow heads show punctate LBD-seeded inclusions. Scale bars are 10 µm. (**C**) Quantification of the aggregate burden as a percentage of the total cells area at 10 days post-treatment with un-amplified as well as SAA-amplified brain homogenates from MSA and LBD patients. The graph shows that MSA is more seeding-competent than LBD and that amplification does not change the seeding properties of brain homogenates. Data represent mean ± SEM of 3 patients, each with 3 images. * *p* < 0.05; One-way ANOVA, Sidak’s post hoc multiple comparisons test.

**Table 1 cells-14-00684-t001:** Demographic information of LBD and MSA patients from whom CSF or brain samples were obtained.

CSF				BH				
S/No	Clinical Diagnosis	Age/Sex	Disease Duration	S/No	Clinical Diagnosis	Brain Regions	Age/Sex	Disease Duration
**1**	LBD	76/M	11	**1**	LBD	ENT	84/F	10
**2**	LBD	74/M	1	**2**	LBD	MB	86/F	8
**3**	LBD	63/M	3	**3**	LBD	MB	87/M	5
**4**	MSA-P	64/M	1	**4**	MSA-C	RFL	61/M	7
**5**	MSA-C	60/M	6	**5**	MSA-P	RFL	60/M	9
**6**	MSA-C	51/M	2.5	**6**	MSA-P	RFL	71/F	10
**7**	Control	59/F	NA	**7**	Control	RFL	97/F	NA
**8**	Control	69/M	NA	**8**	Control	RFL	82/M	NA
**9**	Control	67/M	NA	**9**	Control	RFL	73/F	NA

Note: Age and disease duration are shown in year(s). MSA-P is defined as MSA with Parkinsonism, whereas MSA-C is defined as MSA with cerebellar ataxia. RFL: Right Frontal Lobe; ENT: Entorhinal Cortex; MB: Mid Brain.

## Data Availability

The data supporting the findings of this study are available on request from the corresponding authors.

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
