# Peer review of "Intracellular Inclusions Induced by Patient-Derived and Amplified α-Synuclein Aggregates Are Morphologically Indistinguishable"

_cells, 2025, doi:10.3390/cells14100684_

Round 1
Reviewer 1 Report
Comments and Suggestions for Authors
The study shows that the hypothesised strain specificy of aSyn derived from MSA and LDB brains or CSF is maintained through rounds of SAA in terms of aggegregate morphology in the HEK239T cell line. Amplification of disease relevant aSyn fibrils is a general challenge for many groups in the field, and SAA offers a possible solution to this, making the study highly relevant. However, given the relatively narrow scope of the study, I think it should be a communication rather than a full research article.
Generally the image analysis of aggregate morphology could be improved, there are many different softwares that can generate morphology measures such as roundness, size etc. making this a more quantitative rather than qualitate comparison.
In materials and methods the authors should list the number of technical replicates in the SAA, and whether SAA reaction from all wells, or only the positive, were pool and used for further seeding experiments.
Figure 3 : it is not clear if the aSyn aggregates for the DLB are inside a cell or outside. Please provide a better image. Please provide an image of the controls cells at the same magnification as that for MSA and DLB seeded cells. The controls cells seems to be highly confluent and not in a great shape, or is this due to the image being out of focus?(the comments tor figure 4 - control cells)
Author Response
Comment 1:"The study shows that the hypothesised strain specificy of aSyn derived from MSA and LDB brains or CSF is maintained through rounds of SAA in terms of aggegregate morphology in the HEK239T cell line. Amplification of disease relevant aSyn fibrils is a general challenge for many groups in the field, and SAA offers a possible solution to this, making the study highly relevant. However, given the relatively narrow scope of the study, I think it should be a communication rather than a full research article. "
Response: We appreciate the reviewer’s valuable suggestion. While this manuscript addresses a relatively narrow scope, it focuses on a critical question in a field that continues to debate whether SAA technology preserves the structure of fibrils. The data presented here demonstrate that the seeding properties of SAA-amplified aggregates closely match those of the unamplified aggregates. We believe this finding is of significant importance and merits publication as a research article to inform the scientific community. Given the challenges in detecting seeding activity in samples with extremely low levels, such as cerebrospinal fluid (CSF), we believe that timely publication of these data is crucial. Moreover, this study provides important evidence that aggregates amplified from CSF accurately reflect the behavior of brain-derived aggregates, further supporting the relevance of SAA for research and clinical applications.
Comment 2:"Generally the image analysis of aggregate morphology could be improved, there are many different softwares that can generate morphology measures such as roundness, size etc. making this a more quantitative rather than qualitate comparison. "
Response:
We appreciate the suggested method of quantification and recognize its value. However, the primary objective of this study is to demonstrate the morphological similarity between amplified and unamplified aggregates, as well as those derived from CSF and brain tissue. While quantification can be informative, our focus is not on measuring aggregate abundance but rather on illustrating that aggregates, regardless of their source or amplification status, exhibit consistent morphological characteristics. The distinction between filamentous or wispy morphologies and a dotted appearance is sufficiently pronounced to be qualitatively evident without the need for further quantification. This morphological observation has also been previously reported by Dr. Marc Diamond’s group—the developers of the biosensor cell system—as cited throughout the manuscript.
Comment 3:"In materials and methods the authors should list the number of technical replicates in the SAA, and whether SAA reaction from all wells, or only the positive, were pool and used for further seeding experiments. "
Response: Thank you for pointing this to our attention. We apologize for missing to mention the replicates. This was corrected under the materials and methods, α-Syn seed amplification assay section as well as under Figure 1 legend A, as follows:
α-Syn seed amplification assay
The second rounds of SAA were conducted in triplicates by diluting 100 times the amplified αSyn from a single well of the first round using fresh αSyn monomer.
Figure 1A.
Each aggregation curve represents an individual biological sample performed in triplicates.
Comment 4: "Figure 3 : it is not clear if the aSyn aggregates for the DLB are inside a cell or outside. Please provide a better image. Please provide an image of the controls cells at the same magnification as that for MSA and DLB seeded cells. The controls cells seems to be highly confluent and not in a great shape, or is this due to the image being out of focus?(the comments tor figure 4 - control cells)"
Response: We appreciate the reviewer’s comment and the opportunity to clarify. The localization was assessed using confocal imaging, which allowed us to visualize different cellular layers and confirm that the signal originates from within the cells. The FRET signal further supports this conclusion, as FRET only occurs when alpha-synuclein monomers tagged with YFP and CFP come into close proximity, which would not happen outside the cell. This proximity is only achieved through aggregation, initiated by the seed present in the SAA-LBD patient sample. This intracellular localization is additionally supported by the Airyscan images presented in Figure 5.
Although we have multiple confocal z-stack images supporting this observation, the platform does not allow us to include them all.
Regarding the controls, images taken at the same magnification are shown in Figure 4. For Figure 3, we intentionally used lower magnification to demonstrate that even at a broader field of view, untreated cells do not show aggregation. This underscores that the absence of aggregates is consistent and not limited to a small, highly magnified region.
Reviewer 2 Report
Comments and Suggestions for Authors
The study “Intracellular Inclusions Induced by Patients-Amplified and-Derived α-Synuclein Aggregates are Morphologically Indistinguishable” (cells- 3582917) contains interesting perspectives, as this article investigated the morphological and seeding properties of α-synuclein (α-Syn) aggregates amplified from brain and cerebrospinal fluid (CSF) samples of Lewy Body Disease (LBD) and Multiple System Atrophy (MSA) patients using Seed Amplification Assay (SAA). Which could support the diagnostic potential of CSF-derived aggregates for differentiating synucleinopathies.
My concerns are listed as major, minor, and trivial ones for authors to consider:
Major concerns
- Authors are suggested to clearly articulate the clinical implications of early differentiation between synucleinopathies in the Introduction.
- What method was used to facilitate the uptake of α-syn aggregates from brain and CSF samples of LBD and MSA patients by the biosensor cells?
- Figure 3, in the images of LBD group, it seems like signals of aggregates are coming from a dead cell or a false signal (which may not be true). Similar confusion in figure 5A, on comparison to figure 5B, it appears un-amplified and SAA-amplified samples from MSA aggregates are more detrimental to the cells, hence the signals are coming from dead/dying cells. As the size and morphology is significantly different in both 5A & 5B. Why did authors not used any counter staining to visualize the nucleus and/or cytosol?
- Authors are suggested to change the simple bar graph to scattered dot bar graph to better illustrate the distribution of individual data points and enhance the visual representation of variability.
Minor Comments-
- Improve figure labeling (scale bar line) consistency and clarity in figure 5B.
- What is the scale bar of the pictures in figure 5A?
Author Response
Comment 1:"Authors are suggested to clearly articulate the clinical implications of early differentiation between synucleinopathies in the Introduction."
Response: We would like to thank the reviewer for their valuable feedback. In the revised manuscript, we have highlighted the importance of early differentiation between two major synucleinopathies—Lewy Body Dementia (LBD) and Multiple System Atrophy (MSA). We reference our previous paper that focused on differential diagnosis and emphasize how our current findings enhance our understanding that SAA amplification does not affect the strain properties of LBD and MSA, as indicated by our cell-based studies.
Comment 2:"What method was used to facilitate the uptake of α-syn aggregates from brain and CSF samples of LBD and MSA patients by the biosensor cells?'
Response: All samples used for cell treatment were sonicated to reduce aggregate size to oligomeric or short fibrillar forms, facilitating cellular uptake without the need for transfection reagents or other delivery agents. This methodological detail is outlined in the Materials and Methods section and highlighted in the Discussion as a strength of our approach, particularly in contrast to studies that rely on lipofectamine or similar reagents for aggregate internalization.
Comment 3:"Figure 3, in the images of LBD group, it seems like signals of aggregates are coming from a dead cell or a false signal (which may not be true). Similar confusion in figure 5A, on comparison to figure 5B, it appears un-amplified and SAA-amplified samples from MSA aggregates are more detrimental to the cells, hence the signals are coming from dead/dying cells. As the size and morphology is significantly different in both 5A & 5B. Why did authors not used any counter staining to visualize the nucleus and/or cytosol?"
Response: Thank you for pointing this out. The signal observed cannot be attributed to a false positive, as it is exceptionally bright. The aggregates shown in Figures 3 and 5 are indeed real, as confirmed by the presence of FRET. FRET can only occur if the monomers expressed in the cells—tagged with either YFP or CFP—come into close proximity through aggregation, which is seeded by the patient samples. Therefore, without intracellular aggregation, FRET would not be observed in Figures 2–4.
This conclusion is further supported by longitudinal confocal imaging of the same live cells over a period of 13 days. These daily images clearly show the progressive growth of aggregates, particularly in cells treated with MSA samples.
Regarding the absence of counterstaining: these are live-cell imaging experiments rather than fixed-cell assays. The cells were plated on coverslip chambers and monitored live over the course of 13 days. Media was changed every 48 hours, during which dead cells were naturally removed. Importantly, we did not observe apoptotic blebbing in cells with aggregates, nor did we see aggregates without associated cells. Cells that appeared apoptotic were deliberately excluded from imaging.
Comment 4:"Authors are suggested to change the simple bar graph to scattered dot bar graph to better illustrate the distribution of individual data points and enhance the visual representation of variability."
Response: Thank you for the suggestion. In response, we have revised Figure 5C by converting the graph into a scatter plot to better display the individual data points.
Reviewer 3 Report
Comments and Suggestions for Authors
The submission by Al-Lahham et al describes their A-SYN aggregates detection by seeding assays and a reporter cell line. In 2024 there were almost 2000 publication on A-SYN, and over 500 are related to seeding assays in the past 5 years. Sensitive measurements of A-SYN cannot be accomplished by immuno and FRET-based detection, hence the amplification step from the seeding assay provides a viable solution. However, the main issue of the seeding assay is variability, one way to overcome that is to include a reference such as PFF aliquoted from the same batch stored at -70’C. Another technical hurdle is the limitation of the dye ThT, hopefully other dyes of improved sensitivity and specificity will be emerged soon.
The main point the authors are making is that the amplified products derived from LBD and MSA brains and CSF are different. It says Suppl Table 1 shows the patients demographics but I’m unable to locate where the Table is. My further question is that which part of the brain were they? And were the samples processed and stored in a similar manner? It is interesting that the proteolytic products from LBD and MSA are different – I wonder how they and the amplification profiles look like from PFF ? another interesting expt is to introduce PFF as a spike and see how they look like under a LBD and MSA background.
Is a HEK293 based reporter cell type biologically representative for A-SYN aggregates detection? There are debates on the uptake mechanism – would 293 lack the pathological transporter?
While confocal is the proper imaging approach, there is no description whether stacks and projection were conducted. Going forward, it shall really benefit the field by automating the confocal imaging in order to examine more cells and programme for a quantitative analysis.
Author Response
Comment 1:” It says Suppl Table 1 shows the patients demographics but I’m unable to locate where the Table is.”
Response: We apologize for the oversight regarding the missing table. It has been included in the revised version.
Comment 2:” which part of the brain were they? And were the samples processed and stored in a similar manner? It is interesting that the proteolytic products from LBD and MSA are different – I wonder how they and the amplification profiles look like from PFF ? another interesting expt is to introduce PFF as a spike and see how they look like under a LBD and MSA background.”
Response: We sincerely thank the reviewer for this question and apologize for not including this information in our paper. We have added brain region information in a separate column in Table 1.
In our αSyn-SAA assay, reactions spiked with PFF aggregates aggregated more quickly and produced higher maximum fluorescence than those spiked with LBD samples, regardless of the source. However, the proteolytic profiles of the end products from the PFF-spiked reactions showed four bands that were indistinguishable from those of the LBD-spiked reactions. Additionally, the morphologies of inclusions induced by PFF in synuclein biosensor cells were similar to those treated with LBD-derived aggregates.
Comment 3:” Is a HEK293 based reporter cell type biologically representative for A-SYN aggregates detection? There are debates on the uptake mechanism – would 293 lack the pathological transporter?”
Response: These synuclein biosensor cells were developed by Marc Diamond (Reference 43) and have been widely utilized in Parkinson’s disease (PD) and multiple system atrophy (MSA) research. The system consists of an engineered monoclonal HEK293T cell line that stably expresses full-length α-synuclein carrying the pathogenic A53T mutation, fused to either cyan fluorescent protein (CFP) or yellow fluorescent protein (YFP), enabling FRET-based detection. These cells are capable of internalizing small α-synuclein filaments and oligomers without the need for transfection or other uptake reagents.
Comment 4:” While confocal is the proper imaging approach, there is no description whether stacks and projection were conducted. Going forward, it shall really benefit the field by automating the confocal imaging in order to examine more cells and programme for a quantitative analysis.”
Response: Thank you for your suggested method. Z-stack acquisitions and maximum intensity projections were conducted as part of the analysis. Although automation of confocal imaging is a viable option and could facilitate high-throughput screening, it was not the primary objective of this study.
Round 2
Reviewer 2 Report
Comments and Suggestions for Authors
Thank you for clearing the confusion related to the color signals. I have no further comments.
Reviewer 3 Report
Comments and Suggestions for Authors
authors have addressed all my issues